# Industrial Strategies to Reduce Acrylamide Formation in Californian-Style Green Ripe Olives

**DOI:** 10.3390/foods9091202

**Published:** 2020-08-31

**Authors:** Daniel Martín-Vertedor, Antonio Fernández, Marta Mesías, Manuel Martínez, María Díaz, Elisabet Martín-Tornero

**Affiliations:** 1Technological Institute of Food and Agriculture (CICYTEX-INTAEX), Junta of Extremadura, Avda. Adolfo Suárez s/n, 06007 Badajoz, Spain; tonymagic13@gmail.com; 2Research Institute of Agricultural Resources (INURA), Campus Universitario, Avda. de la Investigación s/n, 06071 Badajoz, Spain; mmcano@unex.es; 3Institute of Food Science, Technology and Nutrition (ICTAN-CSIC), Jose Antonio Novais 10, 28040 Madrid, Spain; mmesias@ictan.csic.es; 4Environmental Engineering Agronomic and Forestry Department, University of Extremadura, Avda. Adolfo Suárez s/n, 06007 Badajoz, Spain; emartintornero@gmail.com; 5Official College of Pharmacists, C/Ramón Albarrán 15, 06002 Badajoz, Spain; mardiamig@gmail.com

**Keywords:** acrylamide reduction, table olives, sterilization, additives

## Abstract

Acrylamide, a compound identified as a probable carcinogen, is generated during the sterilization phase employed during the processing of Californian-style green ripe olives. It is possible to reduce the content of this toxic compound by applying different strategies during the processing of green ripe olives. The influence of different processing conditions on acrylamide content was studied in three olives varieties (“Manzanilla de Sevilla”, “Hojiblanca”, and “Manzanilla Cacereña”). Olives harvested during the yellow–green stage presented higher acrylamide concentrations than green olives. A significant reduction in acrylamide content was observed when olives were washed with water at 25 °C for 45 min (25% reduction) and for 2 h (45% reduction) prior to lye treatment. Stone olives had 21–26% higher acrylamide levels than pitted olives and 42–50% higher levels than sliced olives in the three studied varieties. When calcium chloride (CaCl_2_) was added to the brine and brine sodium chloride (NaCl) increased from 2% to 4%, olives presented higher concentrations of this contaminant. The addition of additives did not affect acrylamide levels when olives were canned without brine. Results from this study are very useful for the table olive industry to identify critical points in the production of Californian-style green ripe olives, thus, helping to control acrylamide formation in this foodstuff.

## 1. Introduction

Table olives are one of the main pickled products prepared throughout the world. Three main processing technologies are used worldwide to produce table olives: Spanish style, Californian-style and natural olives. The “Spanish-style” technique includes debittering the olives by soaking them in diluted lye solutions, washing them to remove excess lye and partial fermentation in brine. In the “Californian-style” method, olives are immersed in lye solutions, with or without air bubbling to cause darkening via oxidation, and then packaged and sterilized using retorts. The “natural olive” technique involves soaking olives in brine where they are subjected to spontaneous fermentation [1,2]. In general, the aim of all of these processing methods is to remove the natural bitterness of this fruit caused by the glucoside oleuropein, improve the sensory characteristics of olives, and ensure that they are safe for consumption [3].

The Californian-style, including Californian-style black ripe olives and Californian-style green ripe olives, is one of the most commonly used procedures, and olives treated according to this technique are the most commercialized. Both types of olives are obtained through very similar processes, however California-style green ripe olives are produced fresh without being stored in brine and are not submitted to air oxidation in tanks, thus avoiding the loss of their green color. The process starts with olives harvested at the green–yellow stage and, as mentioned before, being subjected to successive treatments with diluted NaOH (lye) to remove the natural bitterness. Then, olives are washed several times with water to remove most of the residual lye and lower the pH to 7–8. In the case of green olives, ferrous gluconate is not added for darkening. Finally, olives are canned in mild salt brine and heat sterilized at temperatures >110 °C [4,5,6].

It has been demonstrated that acrylamide is generated during the sterilization phase in California-style green ripe olives. Acrylamide is a toxic compound classified as a probable human carcinogen by the International Agency for Research on Cancer [7]. It is mainly produced as a result of the reaction between the asparagine amino acid and reducing sugars through the Maillard reaction, although different mechanisms appear to be involved in the formation of acrylamide in table olives [5,8]. In 2015, the European Food Safety Authority (EFSA) confirmed that the presence of acrylamide in foods is a public health concern [9]. EFSA considers Californian-style table olives as a potential source of acrylamide since these foods contain similar or even higher levels to those found in other food products such as French fries, cereals, or coffee. However, acrylamide concentrations in table olives can vary widely between different commercial canned black ripe olives [10], with values from 243 to 1349 ng·g^−1^ being recorded. Recently, olives in brine have been included among the foods that the European Commission recommends for monitoring due to the presence of acrylamide [11]. Mechanisms involved in the formation of this contaminant in this foodstuff should, therefore, be investigated.

Casado and Montaño [6] showed only trace or negligible amounts of acrylamide in olives prior to the sterilization process, however, concentrations increased up to 1578 ng·g^−1^ following sterilization. Thus, acrylamide has been associated with high-temperature treatments in Californian-style olives [5,6,10,12,13]. Furthermore, the application of different intensities of thermal sterilization treatments produces modifications in the sensory profile of Californian-style olives, mainly because it contributes a different cooked defect intensity [14]. In addition to temperature, other processing conditions such as storage time, washing of the fruit after oxidation, darkening methods, olive harvest, or the presence of additives can affect the acrylamide concentration [5,6,15]. Specifically, Charoenprasert and Mitchell [5] reported that acrylamide decreased when stored for longer than 30 days. Reduced lye treatment and washing with water also resulted in a higher acrylamide concentration [6], whilst olives processed when exposed to air had higher acrylamide levels than olives processed without air oxidation. This fact explains the higher acrylamide content of Californian-style black ripe olives when compared with Californian-style green ripe olives [16]. The cultivar also has a large influence on the variability of acrylamide levels in this food, with “Manzanilla de Sevilla” being the olive variety with the highest concentrations of this contaminant [6,13,16].

Additives in the brine solution (amino acids, water-soluble vitamins, or sodium sulfite, amongst others) have been described to influence acrylamide formation during the processing of ripe olives. Consequently, some have been tested as inhibitors of acrylamide formation. Casado et al. [17] found that the addition of cysteine (50 mM) caused a 50% reduction in acrylamide content. Lysine, arginine, and glycine also showed a significant decrease, with percentages dropping by a range of between 24% and 27%. López-López et al. [15] reported that proline and sarcosine are the most potent acrylamide inhibitors, whilst glycine, ornithine, taurine, and γ-aminobutyric acid are also effective. Both authors indicated that sodium bisulfite was the major inhibitor of acrylamide formation, with a low impact on sensory quality. However, this additive is currently not permitted by the European regulation for table olives. On the other hand, the absence of calcium chloride in the brine solution appeared to decrease the levels of acrylamide in olives following sterilization [5]. Based on these results, Charoenprasert and Mitchell [5] suggested that modifications to traditional processing methods such as keeping olives in the brine solution for longer periods, decreasing oxygen exposure time, or reducing sterilization temperatures can reduce acrylamide levels in table olives.

Another strategy that has been proved to control acrylamide formation is adding phenolic compounds to the can, such as olive leaf extract mixed with hydroxytyrosol. These constitute food additives and are introduced after the elaboration process and prior to the sterilization phase [10,13,18]. Casado et al. [17] reduced acrylamide in Californian-style olives by adding different natural vegetables with high antioxidant properties. Specifically, blanched garlic reduced acrylamide content by 23%. In contrast, the same authors found that adding to the brine two phenolic compounds, hydroxytyrosol and 3,4-dihydroxyphenyl glycol, which are naturally present in olives does not significantly affect acrylamide levels.

In conclusion, there are several steps during olive manufacturing that can affect acrylamide formation. As far as we know, most research on acrylamide in table olives has been focused on Californian-style black ripe olives and few studies have been conducted on green ripe olives. For this reason, the aim of the present work was to study the effect of processing conditions on acrylamide formation in Californian-style green ripe olives. Aspects considered included the harvest moment, washing prior to the sterilization process, the use of additives, and the form of presentation (stoned, pitted, sliced, or without brine). To our knowledge, this is the first study to focus on this kind of food. Although the acrylamide content of Californian-style green ripe olives is lower than that of black ripe olives, results from the present study could be very useful for controlling acrylamide production during the industrial processes producing green ripe olives.

## 2. Materials and Methods

### 2.1. Samples

Olives (*Olea europaea* L.) of the “Manzanilla de Sevilla”, “Hojiblanca”, and “Manzanilla Cacereña” varieties (as known as “Sevillana”, “Hojiblanca”, and “Cacereña”, respectively) were obtained from a cultivar that collaborates with the “La Orden-CICYTEX” Research Center (Badajoz, Spain). Olive groves were located in the “Vegas Bajas del Guadiana” region. Olives were handpicked in perfect sanitary conditions during the 2018/19 crop season.

The climate of the region can be described as Mediterranean and sees an average annual rainfall of 404 mm. The olive orchard contained 25-year-old olive trees (6 × 6 m^2^). The soil at the experimental orchard was a sandy loam (depth 2 m). About 3500 cm^3^ water/ha was applied (between the 15 May and the 18 November) via linear drip irrigation. Weeds were controlled with post-emergence herbicides and by applying no-tillage conditions.

The three olive varieties were harvested at the green maturation stage (maturity index (MI = 0) according to skin color and flesh evaluations proposed by Uceda and Frías [19]. Olive sampling was carried out in the morning, taking samples randomly from different parts of the central area of the olive tree. After harvesting, olives were immediately introduced into tanks with NaOH (Dirna, Valencia, Spain) at 0.3% (154 kg table olives/225 L NaOH, approximately) and transported to the factory. Following this, olives were processed according to Californian-style green ripe olive techniques [5,16] in a factory located in the Northwest of Spain. Olives were then subjected to lye treatment, immersing the samples in a 2.5% (*w*/*v*) sodium hydroxide solution. This process took place at a room temperature of 25 °C and lasted until NaOH penetrated the pit of the olive. This solution was then removed, and the olives were placed into fresh water. The pH of the olives was neutralized using lactic acid and carbon dioxide gas until the pH value of the final table olives and brine solutions was 4.0. Following this, table olives (150 g) were packed in cans with different presentation formats (stoned olives, pitted olives, and sliced olives). A brine solution containing sodium chloride (NaCl, Valdequímica, Barcelona, Spain) (2% *w*/*v*) was added to the cans. Olive cans did not contain calcium chloride (CaCl_2_). Finally, olive cans were sterilized in an autoclave at 121 ± 3 °C for 30 min. This treatment is equivalent to the cumulative lethality or “F0 value” of 15 min. Thus, we make sure that this is the treatment duration required to reduce microorganisms at a specified temperature to ensure the inactivation of thermo-resistant spoilage bacterial spores [16].

### 2.2. Experimental Design

Five types of industrial-scale experiments were carried out. The diagram of the experimental design is shown in Figure 1. All treatments were administered in quintiplicate. Acrylamide was determined both in olives and brine for experiments 1 and 2, and in olives, brine, and non-liquid olives for experiments 3, 4, and 5.

#### 2.2.1. Experiment #1—Effect of the State of Ripeness of Olive Fruit

Two different maturation indexes (MI) of each olive variety were evaluated. Table olives were harvested during the same olive crop year at the green maturation stage (MI = 0) and at the yellow–green maturation stage (MI = 1). All samples were subjected to the same Californian-style treatment previously described.

#### 2.2.2. Experiment #2—Effect of Washing Prior to Lye Treatment

Different washing treatments were applied to three varieties of table olives (MI = 0) prior to lye treatment: (i) non-washing, (ii) washing in water at 25 °C for 45 min (short washing), and (iii) washing in water at 25 °C for 2 h (long washing). In all cases, wash waters were removed, and table olives were sprayed for 5 min for superficial washing.

#### 2.2.3. Experiment #3—Effect of the Presentation Format of Olives

Prior to packing, three presentation formats of the non-washed table olives obtained from experiment #2 were prepared: (i) stoned olives; (ii) pitted olives; and (iii) sliced olives. A model PSL-51 olive pitting and slicing machine (OFM Food Machinery, Seville, Spain) was used to obtain the different formats.

A batch of olives was put in brine (2% NaCl) while a second batch was canned without brine (non-liquid olives), with the latter being an innovation product. In this way, olives, brine, and non-liquid olives were evaluated in the three different formats (stone, pitted, and sliced olives).

#### 2.2.4. Experiment #4—Effect of CaCl_2_ Addition to the Brine Solution

After preparing the different presentation formats (stone, pitted, and sliced olives, with and without brine), the influence of adding CaCl_2_ (Tetra Chemicals Europe, Helsingborg, Sweden) to the brine solution was studied. The following conditions were evaluated: (i) addition of CaCl_2_ (2 g∙L^−1^) and (ii) no addition of CaCl_2_.

#### 2.2.5. Experiment #5—Effect of Different Concentrations of NaCl

Samples with the different presentation format (stone, pitted, and sliced olives) obtained from experiment #3 were introduced in different brine solutions: (i) with NaCl at 2% and (ii) with NaCl at 4%. CaCl_2_ was not added to the samples.

### 2.3. Acrylamide Analysis in Olives and Brine Solutions

Samples of olives from the different experiments were crushed with a T-18 Basis Ultra-Turrax^®^ Homogenizer device (IKA, Germany) in order to obtain a homogeneous olive paste. This was then stored at −80 °C whilst awaiting acrylamide analysis. Acrylamide was determined as described by Pérez-Nevado et al. [10]. Briefly, 2 g of table olives were stirred in 10 mL of distilled water, centrifuged at 1677 g at 4 °C for 10 min and filtered through a 0.22 µm nylon syringe filter (FILTER-LAB, Barcelona, Spain). Brine solution was only filtered. Filtered supernatants and brines were cleaned up using PCX (200 mg/3 mL) and PRP (60 mg/3 mL) column cartridges (Telos, Kinesis, Australia).

Sample extracts and calibration standards were analyzed on an Agilent Model 1200 Infinity LC high-performance liquid chromatograph (Agilent Technologies, Palo Alto, CA, USA) coupled with an Agilent Technologies triple quadrupole mass spectrometer, as outlined in the method proposed by Fernández et al. [20].

### 2.4. Statistical Analysis

SPSS 18.0 software was used for statistical analysis (SPSS Inc. Chicago, IL, USA). Data were expressed as mean values followed by the standard deviation (SD). One-way analysis of variance (ANOVA) followed by Tukey’s multiple range test was used. The significance level was set at *p* < 0.05.

## 3. Results and Discussion

### 3.1. Effect of the State of Ripeness of Olive Fruit (Experiment #1)

In order to study the influence of different olive maturation indexes on acrylamide formation, levels of the contaminant were measured after submitting the three olive varieties at the green (MI = 0) and yellow–green (MI = 1) stages of maturation to Californian-style green olive processing. Results are shown in Figure 2.

Acrylamide was detected in all varieties studied. The “Sevillana” table olive variety exhibited the highest content of acrylamide, with mean values ranging from 153 ng·g^−1^ at the green stage of maturation (MI = 0) to 203 ng·g^−1^ at the yellow–green stage of maturation (MI = 1). Values ranging from 44 to 105 ng·g^−1^ have been reported by Charoenprasert and Mitchell [5] in “Sevillana” olives elaborated under the same Californian-style green ripe olive process. Lower concentrations were observed in the “Hojiblanca” variety, with mean values ranging from 103 to 144 ng·g^−1^ (33% and 29% lower than “Sevillana” in MI = 0 and MI = 1, respectively). The lowest levels were found in the “Cacereña” variety, with a mean concentration of 57 ng·g^−1^ in MI = 0 and of 76 ng·g^−1^ in MI = 1 (63% lower than “Sevillana” olives and around 45% lower than “Hojiblanca” olives for the two maturation indexes). Higher acrylamide concentrations in the “Sevillana” variety when compared with “Hojiblanca” and “Cacereña” olives have also been described in Californian-style black ripe olives [6,13]. Differences in acrylamide levels between the varieties may be due to the firmness of the fruit and/or the content of acrylamide precursors [6,10,13].

Yellow–green (MI = 1) table olives presented higher acrylamide concentrations than green olives (MI = 0) in the three olive varieties (32%, 39%, and 33% higher for “Sevillana”, “Hojiblanca”, and “Cacereña” olives, respectively) (Figure 2). This suggests that the most advanced stage of ripeness promotes the formation of this contaminant. This can be explained by the changes that occur in the composition and appearance of olives during the olive ripening process. Initially, the olive has an intense green color and becomes more yellowish during the maturation process. During this process, photosynthesis performed by the olive tree generates sugars and other compounds in the olive fruit, in this way increasing acrylamide precursors and, consequently, promoting greater acrylamide formation during sterilization treatment [10,16,21]. In addition, olives in the green–yellow stage contain a lower amount of phenolic compound than they do in the green stage of maturation, decreasing the protection given by these compounds against acrylamide formation [13].

Acrylamide levels in brine were significantly higher than that in olives for both indexes of maturation in the three studied varieties (Figure 2). This fact could be expected due to the diffusion of compounds from one matrix to another when olives are in brine. On the one hand, precursors of acrylamide in the fresh olives could diffuse into surrounding medium during brine storage [5] and, on the other hand, acrylamide generated after sterilization could also enter the brine. Acrylamide is a hydrosoluble and hydrophilic molecule and, therefore, when in aqueous mediums the chemical balance is expected to be displaced from the fruit to the brine. Similar results were found by Pérez-Nevado et al. [10] in Californian-style black ripe olives. These findings corroborate the information included in the recent publication of European Commission Recommendations [11]. This established that the high content of acrylamide in brined-stored food could be linked to the possible presence of acrylamide in brine.

According to our knowledge, this is the first study relating acrylamide formation and the ripeness state of olives. Considering the results, it is recommended that companies producing Californian-style olives acquire the raw fruits as early as possible when still in the earlier maturation state. In other words, the color of the olives should be as close as possible to green. In addition, due to the acrylamide presence in brine, consumers should remove brine before eating olives. It may also be advisable to add water to the olives for a few minutes to wash them and eliminate acrylamide from the brine. In this way, acrylamide exposure from table olives could be reduced.

### 3.2. Effect of Washing Prior to Lye Treatment (Experiment #2)

The acrylamide content of olives subjected to different washing processes prior to lye treatment is shown in Table 1. Significant differences were observed between the three conditions (non-washing, washing in water at 25 °C for 45 min (short washing), and washing in water at 25 °C for 2 h (long washing)). The lowest levels of acrylamide were observed in olives washed for a long time, followed by short-washed olives and, finally, non-washed olives. When compared with non-washed olives, 2 h washes reduced the formation of the contaminant by around 45% in the three varieties, with 45 min washes causing a reduction of around 25%. According to these results, it could be suggested that the cleaning step prior to lye treatment in lipid-rich foods removes acrylamide precursors from the olive fruit, subsequently decreasing contaminant formation. A similar mitigation strategy has been proposed for potatoes, in that it is recommended to wash or even soak fresh tubers before frying in order to reduce acrylamide precursors in the preparation of French fries or chips [11].

When olives were washed, brine also presented lower acrylamide concentrations (25%, 32%, and 22% lower in “Sevillana”, “Hojiblanca”, and “Cacereña” olives, respectively, when olives were washed during 45 min and 41%, 51%, and 36% lower for the same varieties when olives were washed for 2 h, when compared with levels in non-washed olives). This could be due to a lower diffusion of precursors from the fruits to the brine and lower diffusion of acrylamide from the sterilized olives to the brine.

Casado and Montaño [6] showed that olive fruits with a lower acrylamide concentration were those that underwent a washing process twice as long as that of comparison olives. These researchers indicated that long washing (24 h) following lye treatment reduced acrylamide content by approximately 50% in “Sevillana” olives (from 1340 to 680 ng·g^−1^;) and around 80% in the “Hojiblanca” variety (from 1130 to 200 ng·g^−1^). These percentages are much higher than those obtained in the present study, probably due to the shorter washing time employed in our experiment but also due to the step in the process during which washing was carried out. These authors washed olives during the NaOH treatment, which favors greater diffusion and elimination of precursors. It is known that treatments with NaOH modify olive texture, making them less hard and more porous [10,22]. This promotes greater diffusion of compounds from the olives to the brine.

In conclusion, industries should include a washing step prior to the elaboration process in order to reduce acrylamide precursors and, consequently, to decrease contaminant formation. According to the results of the present experiment, washing in water at 25 °C for 2 h could be an adequate mitigation strategy.

### 3.3. Effect of the Presentation Format of Olives on Acrylamide Content (Experiment #3)

The influence of the different table olive presentation formats (stone, pitted, and sliced olives) prior to packing, with and without brine, on acrylamide content in the three varieties is shown in Figure 3. For olives canned in brine, stone olives presented the highest concentrations in the three studied olive varieties, with levels 21–26% greater than pitted olives and 42–50% higher than sliced olives. Stone olives have a compact format enabling a smaller contact surface between the fruit and the liquid. This reduces the diffusion of both acrylamide and acrylamide precursors between the matrices, thus leading to the highest levels. In contrast, sliced olives have a greater contact surface with the brine solution throughout the different steps of the industrial elaboration process. This promotes a greater diffusion of acrylamide and acrylamide precursors from the olives to the brine. This fact explains why sliced presentations showed the lowest concentrations of this toxic substance. Our results agree with those of Casado and Montaño [6], verifying that oxidized sliced black olives pertained to the format with the lowest final acrylamide concentration following the sterilization process. As expected, pitted olives exhibited moderate values for acrylamide levels (Figure 3).

Similarly to previously conducted experiments, brine presented significantly higher concentrations than olives canned in brine and the trend according to the different format presentations was similar to those observed in the fruits: stone > pitted > sliced.

This trend was also noticed in olives canned without brine (non-liquid olives). Again, stone olives displayed the highest acrylamide concentration (20% higher than pitted olives and 45% higher than sliced olives). When compared with olives canned with brine, levels of the contaminant were significantly higher in samples canned without brine (non-liquid olives), with recorded values being as much as three times larger. No previous information has been found regarding the formation of acrylamide in olives canned without brine during the sterilization process. However, it could be deduced that the absence of a liquid medium in the can implies that there is no diffusion, of either the precursors or of acrylamide, to the aqueous medium and, therefore, all the acrylamide generated remains in the olives. In addition, when olives are canned with brine, the heat applied during sterilization is transmitted by convection and the thermal process occurs in a less aggressive way [10]. The absence of liquid could cause overheating in olives, leading to increased formation of acrylamide. For that reason, the consumption of olives with this presentation format, with more than 250 ng g^−1^ of acrylamide, should be controlled, especially among children and elderly consumers, in order to reduce a high acrylamide exposure and, consequently, to prevent possible health risks [5,18].

### 3.4. Effect of CaCl_2_ Addition to the Brine Solution (Experiment #4)

CaCl_2_ is a firming agent, which is frequently added to the covering brine of packed table olives to improve olive firmness [23]. The influence of the addition or absence of CaCl_2_ in the brine solution on acrylamide formation in the different presentation formats of table olives and after being submitted to the sterilization process is shown in Figure 4.

Olives canned with added CaCl_2_ showed acrylamide levels approximately 20% higher than samples without CaCl_2_. These higher levels coincide with the lower acrylamide concentrations of the brine in samples treated with CaCl_2_. These results agree with those reported by Casado et al. [17] in an olive model system with olive juice obtained from “Hojiblanca” olives. They also coincide with those described by Charoenprasert and Mitchell [5] in Californian-style black ripe olives. These authors indicated that, in general, calcium ions help to maintain structural firmness and stability of the cell wall and the cellular turgor of the fruits that form cross links between pectin molecules. This strengthens plant cells and prevents their collapse. It is deduced that calcium reduces the transcription of acrylamide and acrylamide precursors between olives and brine despite these compounds being hydrophilic. The retention of acrylamide precursors inside the fruit promotes a higher formation of the contaminant, whilst the retention of generated acrylamide in sterilized olives increases its concentration in the final product.

The effect of adding calcium to samples canned without brine was not significant and similar concentrations were observed between olives treated and not treated with this additive. This was the case regardless of presentation format and olive variety. In this case, both acrylamide and acrylamide precursors could not migrate from olives to brine in cases where brine was not present. In summary, the presence of CaCl_2_ does not affect acrylamide formation but can reduce the diffusion of acrylamide and acrylamide precursors from olives to brine when olives are canned in an aqueous environment. The addition of this additive to brine should be avoided in order to reduce the acrylamide content of olives after the sterilization process. However, industries must take into account that the firmness of the olives could be modified, and consumer’s acceptance may be affected. For olives without brine, the use of CaCl_2_ can be continued.

### 3.5. Effect of Different NaCl Concentrations (Experiment #5)

The influence of different NaCl concentrations in brine on acrylamide content in the different presentation formats of olives and varieties is shown in Table 2. NaCl is added during the processing of the table olive in order to give a salty taste to the final product. In this experiment, two different NaCl concentrations (2% and 4%) were added to the different olive samples and acrylamide content was analyzed in both the olives and the brine. Acrylamide concentration in olives in the three presentation formats was higher when 4% of NaCl was added to the brine. In contrast, acrylamide decreased in brine with a higher concentration of NaCl. Results are similar to those found in experiment #4 with the application of CaCl_2_ but to a lesser extent. Although the objective of adding NaCl is different, resulting behavior suggests that this additive also indirectly strengthens the olive cellular membranes and, therefore, prevents the diffusion of acrylamide precursors from the olives to the brine. This gives rise to an increase in acrylamide content. In agreement with this finding, Fadda et al. [24] reported that olives canned in brine with higher NaCl percentages showed higher breaking force and hardness values than olives processed with lower NaCl contents.

Regarding the presentation format, when the percentage of NaCl increases, acrylamide formation increases to a lesser extent in stone olives, with greater increases seen in pitted and sliced samples. The greater contact surface in pitted and sliced olives probably allows NaCl to make the cell structures of olives harder, reducing the diffusion of acrylamide precursors from olives to brine and, consequently, increasing acrylamide formation in these formats when compared with stone olives.

Finally, the addition of salt led to no significant differences when olives were not canned with brine. This is an expected result because, as mentioned before, the absence of a liquid medium in the can prevents the diffusion of precursors and acrylamide to the aqueous medium and, as a result, all generated acrylamide remains in the olives. No bibliographic references have been found to confirm or disprove the results obtained in this experiment.

## 4. Conclusions

Californian-style olives are subjected to extreme temperatures during the sterilization process, promoting the formation of acrylamide. Significant amounts of acrylamide have been found in green ripe olives, although in lower concentrations than that in black ripe olives. The present study examined different mitigation strategies with the aim of reducing acrylamide formation in this foodstuff. According to the results obtained, it could be concluded that raw olives should be harvested at the green stage rather than allowing olives to reach later stages of maturity (green–yellow stage). This may mitigate sugar increases and the consequent formation of acrylamide after the sterilization process. Washing with water at 25 °C for at least 45 min but ideally for 2 h prior to lye treatment reduces the levels of acrylamide precursors in olives and, therefore, acrylamide formation during subsequent thermal treatment. Another point to control is the presentation format of olives. In this sense, stone olives have significantly more acrylamide content than sliced and pitted olives, since their smaller contact surface reduces the diffusion of acrylamide precursors from olives to the brine, which would enable greater acrylamide formation. Moreover, the addition of some additives, such as CaCl_2_ and NaCl, commonly used in the elaboration of table olives to provide firmness and a salty taste to the fruit, should be controlled since their use increments acrylamide content in olives. These additives prevent diffusion of acrylamide precursors to the brine, thus increasing formation of the contaminant. This last consideration is not relevant when olives are canned without brine. Finally, due to the high presence of acrylamide in brine, it is recommended to wash olives prior to consumption. Results from this study are very useful for the table olive industry as they will enable it to identify critical points in the production of Californian-style green ripe olives and, in this way, control acrylamide formation in this foodstuff.

## Figures and Tables

**Figure 1 foods-09-01202-f001:**
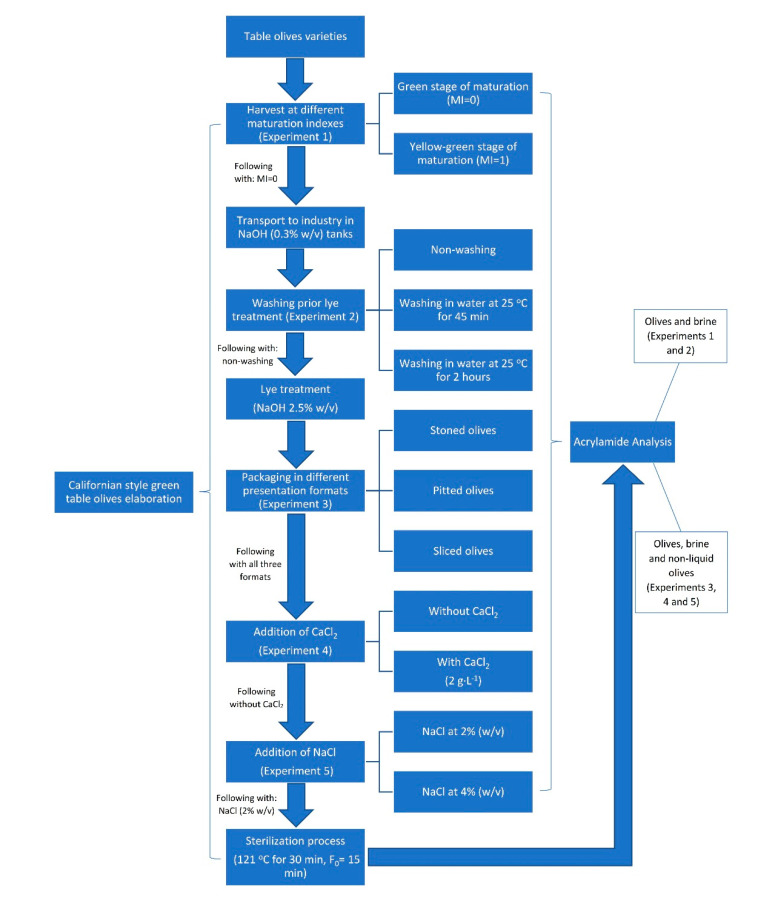
Diagram of the experimental design.

**Figure 2 foods-09-01202-f002:**
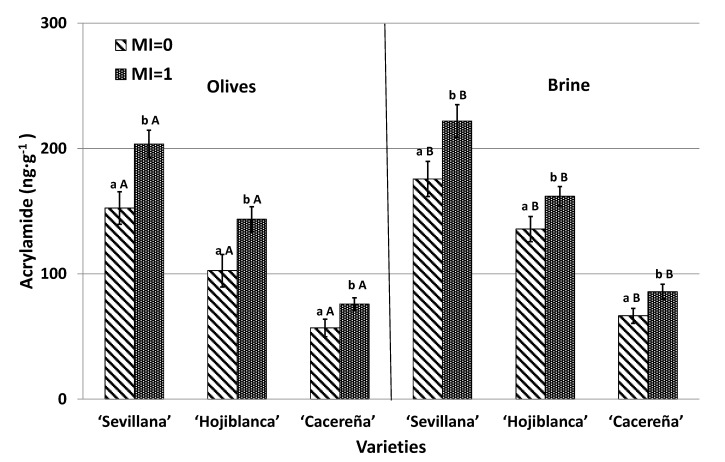
Acrylamide content (ng g^−1^) in olives and brine obtained from different olive varieties harvested at two different olive maturation indexes (MI = 0 and MI = 1) and subjected to Californian-style green table olive processes. Results are expressed as the mean ± SD (standard desviation) of five sample replicates. Superscript indicates significant statistical differences between olives maturation stages for each olive variety (Tukey test, *p* < 0.05). Different capital letters in the same columns indicate significant statistical differences between olives and brine solution for each variety and maturation index (Tukey’s test, *p* < 0.05).

**Figure 3 foods-09-01202-f003:**
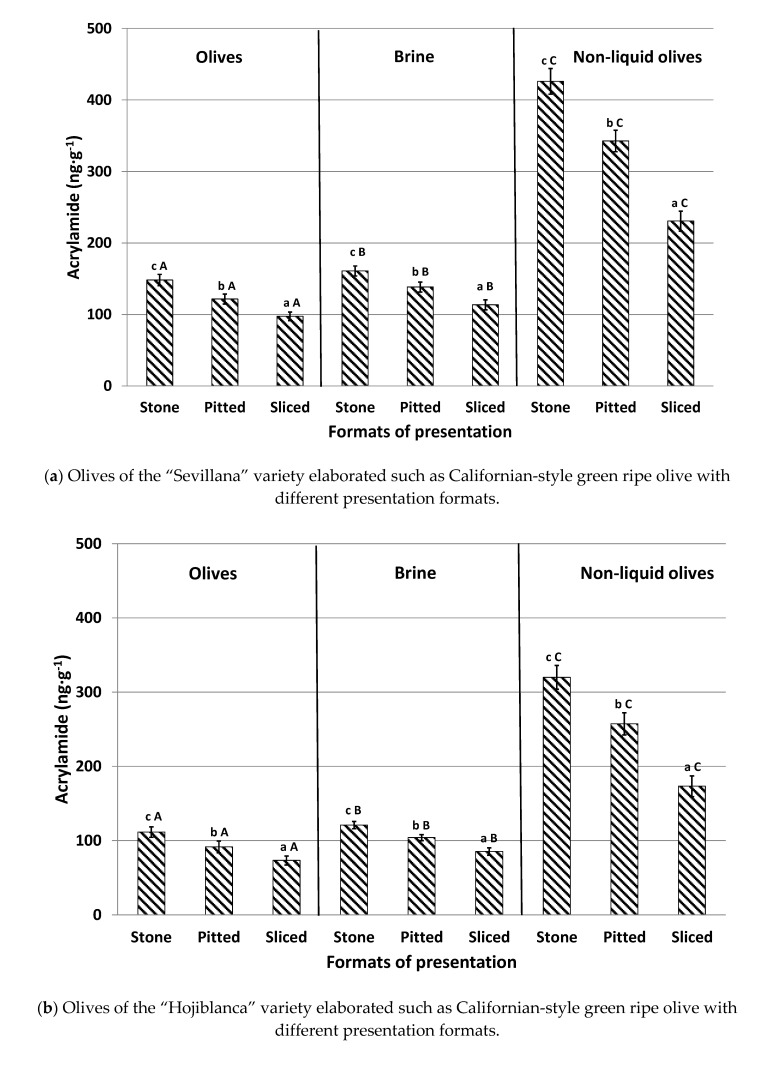
Acrylamide content (ng g^−1^) of olives (maturity index (MI = 0)) and brine with different presentation formats (stone, pitted, and sliced olives), canned with and without brine (non-liquid olives) after the sterilization process. Results are expressed as means ± SD of the five sample replicates. Superscript indicates significant statistical differences between presentation formats in each sample (Tukey test, *p* < 0.05). Different capital letters in the same columns indicate significant statistical differences between raw materials (olives, brine, and non-liquid olives) in each individual presentation format (Tukey´s test, *p* < 0.05).

**Figure 4 foods-09-01202-f004:**
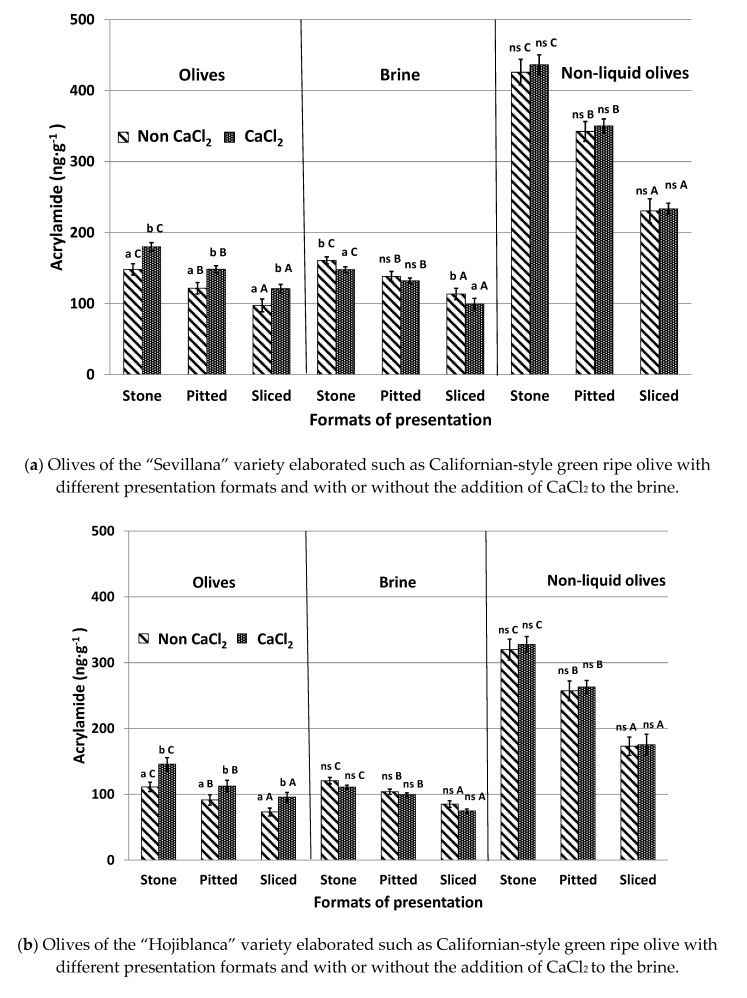
Acrylamide content (ng g^−1^) of olives and brine obtained from different olive presentation formats (stones, pitted, sliced), with or without the addition of CaCl_2_ to brine after the sterilization process. Results are expressed as means ± SD of five sample replicates. Superscript indicates statistically significant differences relating to CaCl_2_ addition (Tukey test, *p* < 0.05). Different capital letters in the same columns indicate statistically significant differences between different olive presentation formats (Tukey test, *p* < 0.05). ns: no significant differences.

**Table 1 foods-09-01202-t001:** Acrylamide content (ng∙g^−1^) of olives and brine after submitting olives to washing (non-washing, washing in water at 25 °C for 45 min (short washing), and washing in water at 25 °C for 2 h (long washing) after the sterilization process. Results are expressed as mean ± SD of five sample replicates. Superscript in the same row indicates statistically significant differences between washing processes (Tukey test, *p* < 0.05). Different capital letters in the same columns indicate statistically significant differences between olives and brine in each treatment (Tukey test, *p* < 0.05).

		Acrylamide (ng∙g^−1^)		
	Samples	Non-Washing	Short Washing	Long Washing
“Sevillana”	Olives	210 ± 8 ^c A^	161 ± 5 ^b A^	124 ± 7 ^a A^
“Sevillana”	Brine	233 ± 9 ^c B^	175 ± 8 ^b B^	137 ± 4 ^a B^
“Hojiblanca”	Olives	151 ± 10 ^c A^	108 ± 7 ^b A^	84 ± 7 ^a A^
“Hojiblanca”	Brine	184 ± 8 ^c B^	125 ± 7 ^b B^	90 ± 7 ^a B^
“Cacereña”	Olives	79 ± 6 ^c A^	58 ± 8 ^b A^	44 ± 8 ^a A^
“Cacereña”	Brine	84 ± 7 ^c B^	65 ± 7 ^b B^	54 ± 4 ^a B^

**Table 2 foods-09-01202-t002:** Acrylamide content (ng∙g^−1^) of olives and brine after NaCl addition to brine at different concentrations (2% and 4%) after the sterilization process. Results are expressed as mean ± SD of the five sample replicates. Superscript in the same row indicates statistically significant differences between NaCl treatments (Tukey test, *p* < 0.05). Different capital letters in the same columns indicate statistically significant differences between olive presentation formats in each variety (olives, brine, and non-liquid olives) and NaCl treatment.

Varieties	Samples		Acrylamide (ng∙g^−1^)	
			2 % NaCl	4 % NaCl
“Sevillana”	Olives	Stone	146 ± 9 ^a C^	166 ± 9 ^b C^
		Pitted	122 ± 10 ^a B^	155 ± 9 ^b B^
		Sliced	97 ± 7 ^a A^	117 ± 7 ^b A^
“Sevillana”	Brine	Stone	161 ± 7 ^b C^	147 ± 11 ^a C^
		Pitted	138 ± 11 ^b B^	110 ± 9 ^a B^
		Sliced	114 ± 8 ^b A^	96 ± 8 ^a A^
“Sevillana”	Non-liquid olives	Stone	426 ± 20 ^ns C^	431 ± 13 ^ns C^
		Pitted	343 ± 22 ^ns B^	359 ± 20 ^ns B^
		Sliced	231 ± 22 ^ns A^	223 ± 20 ^ns A^
“Hojiblanca”	Olives	Stone	111 ± 9 ^a C^	137 ± 9 ^b C^
		Pitted	91 ± 10 ^a B^	115 ± 9 ^b B^
		Sliced	73 ± 7 ^a A^	97 ± 7 ^b A^
“Hojiblanca”	Brine	Stone	120 ± 7 ^b C^	107 ± 9 ^a C^
		Pitted	101 ± 8 ^b B^	90 ± 9 ^a B^
		Sliced	85 ± 8 ^b A^	73 ± 8 ^a A^
“Hojiblanca”	Non-liquid olives	Stone	320 ± 20 ^ns C^	331 ± 13 ^ns C^
		Pitted	257 ± 22 ^ns B^	269 ± 20 ^ns B^
		Sliced	173 ± 21 ^ns A^	163 ± 20 ^ns A^
“Cacereña”	Olives	Stone	64 ± 5 ^a C^	87 ± 6 ^b C^
		Pitted	52 ± 5 ^a B^	75 ± 6 ^b B^
		Sliced	42 ± 6 ^a A^	67 ± 6 ^b A^
“Cacereña”	Brine	Stone	69 ± 7 ^b C^	60 ± 7 ^a B^
		Pitted	59 ± 7 ^b B^	40 ± 6 ^a AB^
		Sliced	49 ± 5 ^b A^	41 ± 7 ^a A^
“Cacereña”	Non-liquid olives	Stone	183 ± 20 ^ns C^	171 ± 13 ^ns C^
		Pitted	147 ± 22 ^ns B^	159 ± 20 ^ns B^
		Sliced	99 ± 22 ^ns A^	103 ± 20 ^ns A^

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
