# Peer review of "Industrial Strategies to Reduce Acrylamide Formation in Californian-Style Green Ripe Olives"

_foods, 2020, doi:10.3390/foods9091202_

Round 1
Reviewer 1 Report
The authors provide an excellent study of an industrial-scale experiment to study the influences of acrylamide formation during processing of certain table olives. The methods are described in detail, the results are statistically evaluated and the conclusions are based in the available data.
I have not much to criticize apart from a question regarding sensory testing of the olives. Do the suggested measures to reduce acrylamide lead to olives, which are completely different in profile? Would the consumer still buy such olives? This is potentially a topic for follow-up research.
Author Response
The authors provide an excellent study of an industrial-scale experiment to study the influences of acrylamide formation during processing of certain table olives. The methods are described in detail, the results are statistically evaluated and the conclusions are based in the available data.
-We thank the Reviewer for the positive and kind remarks regarding our work. We have revised the different reviewer´s suggestion point by point.
I have not much to criticize apart from a question regarding sensory testing of the olives. Do the suggested measures to reduce acrylamide lead to olives, which are completely different in profile?
-Applying the different strategies in the industry, it is possible to reduce the acrylamide content in the olive varieties studied. Thus, the acrylamide profile is completely different to those control olives without the application of industrial strategies. Our results are quite robust since they were studied in three different olive varieties, obtaining the same trend in the acrylamide synthesis.
-Moreover the olives obtained with these modifications during the production process of Californian-style green ripe olives lead to olives with not different sensory profile than the control olives. Researchers have showed that the application of different intensities of thermal sterilization treatments produces a different sensory profile in the olives, mainly because it contributes a different cooked sensory defect [14]. However, in our study, we have applied the same sterilization treatment in all the experiments done (See the new paragraph added: lines 170-174, page 8).
-Following the reviewer’s comment, we have included a new paragraph in the introduction part in order to clarify this aspect (See lines 87-90, page 4).
Would the consumer still buy such olives? This is potentially a topic for follow-up research.
-We completely agree with the Referee, this topic is an important way for future research.
-We have added a new paragraph (Lines 354-357, page 16) explaining that this kind of olive, overall non-liquid olives, produced high amount of acrylamide and therefore the consumption of high amounts of these olives could be linked to possible toxicological effects. In this way this article proposes new strategies to reduce the acrylamide content in the final product to be safer for the consumer.
Reviewer 2 Report
Industrial strategies to reduce the acrylamide content of foods are an important issue. The manuscript is dealing the same for table olive. I have found the manuscript well prepared, followed the standard methodologies, and obtained reasonable results. Here are my observations,
- Please write degree Celcius correctly (remove '_' from the below of degree sign)
- Figure 2,3,4: Please show SD (standard deviation) bar for every Figure.
- In the experimental design flowchart (Figure 1) following the washing treatment (prior lye i.e. 0.3% NaOH treatment) there is another lye treatment with 2.5% w/v NaOH. Why this step has not been performed in the experiment.
- Page 10: Please cite reference for the mechanism of modification of olive texture by NaOH (like mentioned on age 10: '...... NaOH........making then less hard and more porous'.'
- Figure 4: Non-liquid olives contained a higher amount of acrylamide (which is even higher than that was present in green and yellow-green olive). This means somehow processing (olive was transported to the industry in 0.3% NaOH, washing/non-washing, lye treatment and addition of calcium chloride) influences to develop acrylamide (from amino acids, etc. probably) although no heat was applied. Please explain this part and cite suitable references.
- Figure 4: Please write CaCl2.
- Page 15: Please correctly write NaCl (not NaCL).
- Since 1980 microbial degradation of acrylamide has also been explored. I suggest authors may mention if this method is being used in the olive processing industry.
Author Response
Industrial strategies to reduce the acrylamide content of foods are an important issue. The manuscript is dealing the same for table olive. I have found the manuscript well prepared, followed the standard methodologies, and obtained reasonable results. Here are my observations,
-We would like to thank the referee for the useful comments
- Please write degree Celsius correctly (remove '_' from the below of degree sign)
-We have written degree Celsius correctly through the manuscript.
- Figure 2, 3, 4: Please show SD (standard deviation) bar for every Figure.
-The standard deviation has been included in all the figures of the manuscript.
- In the experimental design flowchart (Figure 1) following the washing treatment (prior lye i.e. 0.3% NaOH treatment) there is another lye treatment with 2.5% w/v NaOH. Why this step has not been performed in the experiment.
-This fact is included in the material and method epigraph.
-We consider that the flowchart is correct because all the experiments follow that sequential flow.
-Olives in the different experiments are submitted to a final lye treatment (2.5% w/v NaOH) to eliminate the olives bitterness. This fact is explained in Lines 157-162, page 7.
- Page 10: Please cite reference for the mechanism of modification of olive texture by NaOH (like mentioned on page 10: '...... NaOH........making then less hard and more porous'.'
-Thank you for that consideration. We have included a new reference in this sentence. See line 316, page 14.
- Figure 4: Non-liquid olives contained a higher amount of acrylamide (which is even higher than that was present in green and yellow-green olive). This means somehow processing (olive was transported to the industry in 0.3% NaOH, washing/non-washing, lye treatment and addition of calcium chloride) influences to develop acrylamide (from amino acids, etc. probably) although no heat was applied. Please explain this part and cite suitable references.
-The Introduction part explains that acrylamide is only synthesized after submitted olives to sterilization treatments. For example, see lines 84-86, page 4.
-Besides, we have clarified this issue through the manuscript. See lines 361-365, page 16 and lines 386-387, page 17.
-This fact is also included in material and method part (lines 170-174, page 8).
- Figure 4: Please write CaCl2.
-This fact has been done. See Figure 4.
- Page 15: Please correctly write NaCl (not NaCL).
-This fact has been done. See lines 406, page 18.
- Since 1980 microbial degradation of acrylamide has also been explored. I suggest authors may mention if this method is being used in the olive processing industry.
-This seems very interesting in other foods, however, in the oxidized or non-oxidized olives, the elaboration process finishes with a thermal sterilization treatment (approximately 121℃ during 30 min), which causes the elimination of all vegetative forms and spores, so it would not be feasible to add microorganisms to reduce this toxic substance.
Reviewer 3 Report
Comments and Suggestions for Authors

Author Response
Introduction section
- P4 Line 20 Please revise “ ….Green ripe olives. ” to “ ….green ripe olives. ”
-This fact has been done. See line 131, page 6.
Materials and Methods section
- P5 Line 20 line 5 Please explain the noun “F0”.
-Thank you for that consideration. This fact has been clarified in the manuscript. See lines 170-174, page 8.
- P6 Figure 1 Some words are hard to be recognized in the figure. Please improve the resolution.
-This fact has been done. Thank you. The figures have been inserted at the highest resolution in which it can be uploaded to the Journal.
Results and Discussion
- P13 Figure 4 Please revise “ CaCl2 ” to “CaCl2 ”
-This fact has been done. See Figure 4.
- P15 Line 6 and Line 6 Please revise “ NaCL” to “NaCl ”
-This fact has been done. See lines 406, page 18.